# The Association between Anemia and Parkinson’s Disease: A Nested Case-Control Study Using a National Health Screening Cohort

**DOI:** 10.3390/brainsci11050623

**Published:** 2021-05-13

**Authors:** Ji Hee Kim, Jae Keun Oh, Jee Hye Wee, Chan Yang Min, Dae Myoung Yoo, Hyo Geun Choi

**Affiliations:** 1Department of Neurosurgery, Hallym University College of Medicine, Anyang 14068, Korea; kimjihee.ns@gmail.com (J.H.K.); ohjaekeun@gmail.com (J.K.O.); 2Department of Otorhinolaryngology, Hallym University College of Medicine, Anyang 14068, Korea; weejh07@hanmail.net; 3Hallym Data Science Laboratory, Hallym University College of Medicine, Anyang 14068, Korea; joicemin@naver.com (C.Y.M.); ydm1285@naver.com (D.M.Y.); 4Hallym Data Science Laboratory, Department of Otorhinolaryngology-Head & Neck Surgery, Hallym University College of Medicine, Anyang 14068, Korea

**Keywords:** anemia, hemoglobin, iron, neurodegeneration, Parkinson’s disease, pathogenesis

## Abstract

(1) Background: Controversy exists regarding the relationship between anemia and Parkinson’s disease (PD). This study aimed to evaluate the risk of PD related to anemia in the Korean population. (2) Methods: The Korean National Health Insurance Service-National Sample Cohort, which includes adults over 40 years of age, was assessed from 2002 to 2015. A total of 5844 PD patients were matched by age, sex, income, and region of residence with 23,376 control participants at a ratio of 1:4. The analyzed covariates included age, sex, blood pressure, fasting blood glucose, obesity, smoking status, and alcohol consumption. A multiple logistic regression analysis was conducted for case-control analyses. (3) Results: The adjusted odds ratio (OR) for the risk of PD associated with anemia was 1.09 after adjusting for potential confounders (95% confidence interval (CI) 1.01–1.18, *p* = 0.030). Among men younger than 70 years, the adjusted OR of PD was 1.34 (95% CI 1.13–1.60, *p* = 0.001). (4) Conclusions: Our findings suggest that anemia may increase the risk of PD, particularly in men younger than 70 years. Further research is required to elucidate the causal relationship between these two diseases.

## 1. Introduction

Anemia is a common condition in which hemoglobin concentration and/or erythrocyte (red blood cell, RBC) levels are lower than normal and not enough to satisfy an individual’s physiological requirements [1]. Anemia remains a highly prevalent global health issue, affecting almost one-quarter of the world’s population according to the World Health Organization (WHO), with 50% of circumstances attributable to iron deficiency [2]. Previous research has found that anemia contributes to morbidity, mortality, poor physical performance, and disability [3,4,5,6,7,8]. Major health consequences of anemia include an increased risk of maternal and child mortality, impaired cognitive and physical development in children, decreased physical function and work efficiency in adults, and cognitive impairment in elderly individuals [2,9].

Parkinson’s disease (PD) is the second most prevalent and complex neurodegenerative disorder and affects more than seven million people worldwide. PD is specified by early noticeable death of dopamine-containing neurons in the substantia nigra pars compacta. The subsequent dopamine deficiency in the basal ganglia causes classic parkinsonian motor symptoms including resting tremor, rigidity, and bradykinesia. The α-synuclein gene encodes a presynaptic protein that has a tendency to misfold and is consequently found to be a key constituent of Lewy bodies, which is a crucial pathologic feature of the PD brain [10]. In addition to motor symptoms, nonmotor characteristics are commonly seen in PD patients earlier than the onset of classic motor presentation. This premotor or prodromal stage of the disease can be characterized by olfactory dysfunction, constipation, depression, excessive daytime sleepiness, and rapid eye movement sleep behavior disorder (RBD) [11].

Evidence is accumulating regarding the link between hemoglobin levels and PD risk. Numerous previous studies have described that individuals with lower hemoglobin levels are more vulnerable to PD and that anemia is possibly related to the development of PD [12,13,14]. Moreover, a case-control study in the U.S. suggested that anemia can precede the onset of PD as a prodromal phase of PD [15]. However, one study reported that a surge in hemoglobin concentration in peripheral blood was correlated with an increased risk of PD in elderly men [16], and another recent study revealed that lower hemoglobin numbers were negatively connected with the risk of development of PD [17]. To date, the relationship between anemia or the level of hemoglobin and PD has not been fully elucidated.

Therefore, the current study aimed to examine the association between anemia and PD after adjusting for potential confounders, including a variety of comorbidities. The second objective was to explore whether these associations were significantly different among the groups in a subgroup analysis stratified by age, sex, and several risk factors.

## 2. Materials and Methods

### 2.1. Study Population

This study was approved by the ethics committee of Hallym University (23 October 2019). The requirement for written informed consent was waived by the Institutional Review Board. All analyses followed to the guidelines and regulations of the ethics committee of Hallym University. All-inclusive characteristics of the Korean National Health Insurance Service (NHIS)-Health Screening Cohort data are presented elsewhere [18].

### 2.2. Definition of Anemia (Independent Variable)

Among the health checkup data, the most recent hemoglobin concentration before the diagnosis of PD (index date) was used. Anemia was indicated by hemoglobin levels <13 g/dL in men and <12 g/dL in women [19].

### 2.3. Definition of PD (Dependent Variable)

PD was defined if the participants were diagnosed with International Classification of Diseases and Related Health Problems (ICD)-10 code G20 (Parkinson’s disease). For the accuracy of the diagnosis, we included individuals who had visited hospitals or clinics ≥ 2 times for the diagnosis of PD following our previous study [20].

### 2.4. Participant Selection

PD patients were included from 514,866 participants with 615,488,428 medical claim codes from 2002 to 2015 (n = 6483). Participants were included in the control group if they were not defined as having PD from 2002 to 2015 (n = 508,383). To sort PD patients who were diagnosed for the first time, we excluded PD patients who were diagnosed between 2002 and 2003 (washout period, n = 637). Participants who did not have a hemoglobin record before the day of PD treatment were excluded (n = 2 for PD patients, n = 99 for control participants). PD patients were 1:4 matched with control participants for age, sex, income, and region of residence. To reduce selection bias, the control participants were assigned in random number order. The index date of each PD patient was fixed as the time of PD treatment initiation. The index date of control participants was arranged as the index date of their matched PD patients. Consequently, every PD patient and matched control participants had the same index date. Throughout the matching process, 484,908 control participants were excluded. Lastly, 5844 PD patients were 1:4 matched with 23,376 control participants (Figure 1).

### 2.5. Covariates

Age groups were designated into 5-year intervals: 40–44, 45–49, 50–54…, and 85+ years old. A total of 10 age groups were determined. Income groups were specified as 5 classes (class 1 (lowest income)–5 (highest income)). The regions of residence were grouped into urban (Seoul, Busan, Daegu, Incheon, Gwangju, Daejeon, and Ulsan) and rural (Gyeonggi, Gangwon, Chungcheongbuk, Chungcheongnam, Jeollabuk, Jeollanam, Gyeongsangbuk, Gyeongsangnam, and Jeju) areas.

Smoking status was divided according to the participant’s current smoking status (nonsmoker, past smoker, and current smoker). Alcohol consumption was classified in accordance with the frequency of alcohol consumption (<1 time a week and ≥1 time a week). Obesity was quantified using body mass index (BMI, kg/m^2^). BMI was categorized as <18.5 (underweight), ≥18.5 to <23 (normal), ≥23 to <25 (overweight), ≥25 to <30 (obese I), and ≥30 (obese II) based on the Asia-Pacific criteria according to the Western Pacific Regional Office (WPRO) 2000 [21]. Systolic blood pressure, diastolic blood pressure, and fasting blood glucose were measured. Missing BMI (20/29,200 (0.068%)), systolic blood pressure (16/29,200 (0.055%)), diastolic blood pressure (16/29,200 (0.055%)), and fasting blood glucose (1/29,200 (0.003%)) values were substituted by the mean values of each variable from the final group of included participants. The Charlson Comorbidity Index (CCI) measuring disease burden by 17 comorbidities was used as a continuous variable (0 (no comorbidities) through 29 (multiple comorbidities)) [22,23]. In the current study, we excluded cerebrovascular disease and dementia from the CCI score.

Regarding PD, head trauma history (ICD-10 codes S00 to S09, diagnosed by neurologists, neurosurgeons, or emergency medicine doctors) in individuals who experienced head and neck computed tomography (CT) evaluations (claim codes: HA401-HA416, HA441-HA443, HA451-HA453, HA461-HA463, or HA471-HA473) and other degenerative diseases of the nervous system (ICD-10 codes G30 to G32, diagnosed by neurologists) were additionally included as covariates if participants had treatment ≥ 2 times.

### 2.6. Statistical Analyses

The overall characteristics of the PD and control groups were compared by means of the chi-squared test.

To obtain the odds ratios (ORs) with 95% confidence intervals (CIs), a conditional logistic regression analysis was performed for the PD and control groups. In these analyses, the unadjusted model and the model adjusted for multiple covariates including obesity, smoking, alcohol consumption, systolic blood pressure, diastolic blood pressure, fasting blood glucose, CCI scores, other degenerative diseases of the nervous system, and head trauma history were considered. The analysis was stratified by age, sex, income, and region of residence. For the subgroup analyses, we divided participants by age and sex (<70 years old and ≥70 years old; men and women) and examined the unadjusted and adjusted models.

In another subgroup analysis, we analyzed the participants depending on obesity, smoking, alcohol consumption, blood pressure, and fasting blood glucose. In that analysis, we analyzed participants in a nonstratified adjusted logistic regression model.

Two-tailed analyses were achieved, and significance was indicated by *p* values less than 0.05. SAS version 9.4 (SAS Institute Inc., Cary, NC, USA) was utilized for statistical analyses.

## 3. Results

The general characteristics (age, sex, income, and region of residence) of participants were the same between the two groups due to the matching procedures (*p* = 1.000; Table 1). The rates of smoking, alcohol consumption, fasting blood glucose, CCI score, other degenerative diseases of the nervous system, and head trauma history significantly differed between the PD and control groups (*p* < 0.001; Table 1).

The unadjusted OR, which was adjusted only for age, sex, income, and region of residence, for anemia in the PD group was 1.13 (95% CI = 1.05–1.23; *p* < 0.001; Table 2). This result was consistently significant; the adjusted OR was 1.09 (95% CI = 1.01–1.18; *p* = 0.030) after adjusting for potential confounders, including comorbidities.

In subgroup analyses stratified by age and sex, the adjusted OR for anemia was significantly higher in the PD group among men younger than 70 years (Table 2). In addition, the adjusted OR for anemia was higher in the PD group among individuals with a BMI ≥23 when subgroup analyses were conducted according to BMI, and the adjusted OR for anemia was higher among individuals with a fasting blood glucose level ≥100 mg/dL when subgroup analyses were implemented according to fasting blood glucose level (Appendix A, Figure 2).

## 4. Discussion

This nested case-control study investigated the association between anemia and PD using an age-, sex-, income-, and region of residence-matched cohort. We found that individuals with PD had a higher prevalence of diagnosed anemia than individuals without PD. The adjusted OR for anemia in the PD group was 1.09 (95% CI = 1.01–1.18) in comparison with the control group after adjusting for age, sex, socioeconomic factors, and comorbidities. This result was in line with those of the subgroup of men younger than 70 years. These findings are similar to those of a cohort in Israel showing that anemia was associated with significantly higher PD risk among men, with an age-pooled hazard ratio of 1.19; an especially high risk was identified among individuals between the ages of 60 and 64 years. Furthermore, our finding showed that this association persisted among individuals with a BMI ≥ 23 and a fasting blood glucose of 100 or higher when it was stratified by fasting blood glucose and BMI.

The pathophysiological relationship between anemia and PD is still controversial, and there are many hypotheses explaining this relationship. As a mechanism elucidating the link between anemia and dementia, one possible explanation is that directly impaired brain oxygenation caused by a chronic anemic state disrupts neural activity [7,24]. This is because a chronic anemic state leads to a reduced capacity of erythrocytes to deliver sufficient oxygen to body cells and tissues to maintain cell viability.

Another explanation is that iron metabolism dysfunction can contribute to the development of PD because iron deficiency impairs erythropoiesis and enhances eryptosis and suicidal erythrocyte death, resulting in anemia. Basically, iron has a crucial role in dopaminergic cell development as a significant cofactor of tyrosine hydroxylase, which is accompanied in dopamine biosynthesis, and of monoamine oxidase (MAO), an enzyme that is involved in dopamine metabolism [25]. Therefore, disrupted iron homeostasis, with both insufficient or high levels of iron, can lead to a variety of neurodegenerative diseases, including PD. One preclinical study reported that for iron regulatory protein, which regulates cellular iron homeostasis, knockout mice developed progressive neurodegeneration [26]. Additionally, rats served an iron-restricted diet developed an obvious reduction in striatal dopamine and decreased dopaminergic activity, and iron deficiency impaired dopamine reuptake in a mouse model [27,28]. A recent meta-analysis observed a substantial downregulation of genes related to hemoglobin and iron metabolism, such as hemoglobin delta, hemoglobin stabilizing protein, and solute carrier family 11 membrane 2, in 4 blood microarrays of PD patients [29]. Some clinical and epidemiological studies have shown that raised serum iron levels are related to decreased PD risk and that abundant dietary iron intake reduces PD risk [30,31]. However, there is evidence demonstrating that anemia decreases the risk of PD. One study reported that participants with a high intake of iron had an increased risk of PD (OR = 1.7; 95% CI = 1.0–2.7) [32]. A recent population-based study reported that anemia was associated with a lower risk of PD (adjusted hazard ratio = 0.894; 95% CI = 0.809–0.989), particularly among patients with moderate to severe anemia [17]. They reported that the association of anemia and the occurrence of PD can be clarified by oxidative stress, α-synucleinopathy, and dysregulation of iron homeostasis. This is because iron is incorporated in the reactive oxygen species (ROS) system, which results in free radical and oxidative damage to neurons. Furthermore, decreased α-synuclein in the erythrocytes of anemic subjects may lead to decreased secretion of α-synuclein, and thus, the extent of α-synuclein experiencing pathologic change. Considering preclinical and clinical studies comprehensively, both inadequate and excessive iron resulting from dysregulation of iron homeostasis may be related to the pathogenesis of PD, as mentioned above.

The final explanation is the hypothesis that PD and anemia share a common etiology such as oxidative stress. It is well documented that oxidative stress plays a major role in the pathogenesis of PD. Although the exact mechanisms that cause the degeneration of neurons and the involved pathological consequences remain unclear, the central role of oxidative stress in the pathogenesis of PD is connected with triggering a cascade of events, comprising mitochondrial dysfunction, injury of nuclear and mitochondrial DNA, and neuroinflammation, which in turn result in increased ROS production [33]. Likewise, oxidative stress is known to be a positive contributor to anemia, given its effects on lipid peroxidation and DNA damage [34]. A recent study showed that oxidative stress is a chief perpetrator in aggravating erythrocyte damage in various system environments and contributes to the development of anemia in these pathologic states [35]. Erythrocytes are a primary target for oxidative stress because of their principal function as O_2_-carrying cells. Accelerated suicidal erythrocyte death leads to enhanced clearance of iron-deficient erythrocytes, as a result of compounding anemia [36,37]. Accordingly, accumulating evidence suggests that amplified oxidative stress may contribute to the pathogenesis of patients with iron deficiency anemia [38,39,40,41,42,43].

Moreover, anemia per se has noteworthy consequences for human health. The negative effects of anemia on health and developmental outcomes arise from the influences of reduced oxygen delivery to tissue, which may affect several organ systems, as well as effects associated with the underlying causes of anemia, which are problematic to disentangle. Patients diagnosed with anemia usually have a variety of chronic concomitant diseases, and anemia can extensively contribute to the development or aggravation of various diseases. Consequently, it is not surprising that patients diagnosed with PD were more common in the anemic group. Although we adjusted for various comorbidities using the CCI score considering this characteristic of anemia in this study, it can be difficult to correct for the numerous causes of anemia and its systemic effects.

One notable finding of this study is that men with anemia were found to have a significantly higher PD prevalence than those without anemia, but this finding was not observed among women. Generally, anemia is more frequent in women than in men and could be related to gynecological conditions. Although we adjusted for various compounding factors, specific information on those medical conditions was not considered in the analysis. Men may be more vulnerable than women to some of the factors contributing to the pathogenesis of PD described above, such as dysregulation of iron homeostasis, and further investigation of this mechanism might help elucidate the pathogenesis of PD.

Our study has a number of strengths. We used the NHIS data sample, which included participants who represent the whole Korean population. The participants were checked up without missing data. Due to the considerable number of participants, we had the opportunity to randomly choose a control group using 1:4 matching by age, sex, income, and region of residence. Furthermore, we adjusted our analyses for all potential confounders, including several well-known risk and protective factors of PD, such as alcohol consumption, smoking status, BMI, blood pressure, blood glucose, rural living, and traumatic brain injury. We also used stratified analyses by sex, age, obesity, and blood glucose status, which allowed investigation of the modifying effect of these factors on PD risk associated with anemia in a powered analysis.

However, several limitations of this study should be mentioned. First, a clear causal relationship between anemia and PD cannot be proven in this study, even though our study disclosed a significant association between the two diseases. Further investigations—in particular, prospective randomized controlled studies—are warranted to indicate causal associations between anemia and PD. Second, we assessed the presence and severity of anemia according to the hemoglobin level obtained at one time point; thus, we could not define the physiological onset and duration of anemia. Additionally, anemia indicators, such as hematocrit and mean cell volume, and parameters of iron metabolism, such as ferritin, serum iron, and total iron binding capacity, were not available in the Korean NIHS database. Multifactorial etiology is a characteristic of decreased hemoglobin levels. Consequently, only the reduced erythrocyte count or hemoglobin concentration may not be enough to determine the cause of anemia. Although mean cell hemoglobin is an available laboratory marker of iron status, serum ferritin measurement is the most specific test that correlates with total body iron stores. Further studies assessing these hematological indicators are needed to better explain the function of iron metabolism and hemoglobin in contributing to PD risk. In addition, further subgroup analysis regarding the relation between the subtypes of anemia and PD is required to understand the precise mechanism of the association between anemia and PD. Third, information on confounders, such as ethnicity, occupation, diet, physical activity, coffee or caffeine intake, genetic factors, exposure to pesticides and other environmental chemicals, and use of drugs including calcium channel blockers and nonsteroidal anti-inflammatory drugs (NSAIDs), which are putative risk factors for the development of PD [44,45], was not available for analysis in this study. It would be worthwhile to examine the effect of hemoglobin on PD risk adjusted for those potential confounding factors. Fourth, the reliability of PD diagnosis was one concern. The validation was performed by medical review, which is not the standard approach for diagnosing PD in most studies. In fact, no reliable objective test for PD is currently available; hence, expert judgement remains the gold standard for diagnosis among living patients. Accordingly, the misclassification of PD in our database as other parkinsonism or drug-induced parkinsonism may bias the true risk of anemia in the context of PD development. Last, as mentioned earlier, diagnosis of PD occurs with the onset of motor symptoms but may be preceded by a premotor or prodromal phase ranging from years to decades. Thus, we could not rule out a situation in which PD had developed and the pathological process had progressed before the diagnosis of anemia. One study determined that de novo anemic patients might develop PD four or more years after the first diagnosis of anemia [13]. However, another study suggested that anemia in early life can precede the diagnosis of PD by 20 years or longer [15]. Thus, we may interpret anemia as a result of PD rather than as a risk factor for PD.

## 5. Conclusions

We demonstrated that anemia was significantly associated with PD. In subgroup analyses, the relationship between these two diseases was maintained in the group of men younger than 70 years. Our results may add to the growing evidence in support of the suggestion that individuals with lower hemoglobin levels have a higher risk of PD. This association could be explained by impaired brain oxygenation, dysregulation of iron homeostasis, and oxidative stress. Further well-designed cohort studies are essential to clarify the causal relationship and understand what is underlying the pathophysiologic mechanism.

## Figures and Tables

**Figure 1 brainsci-11-00623-f001:**
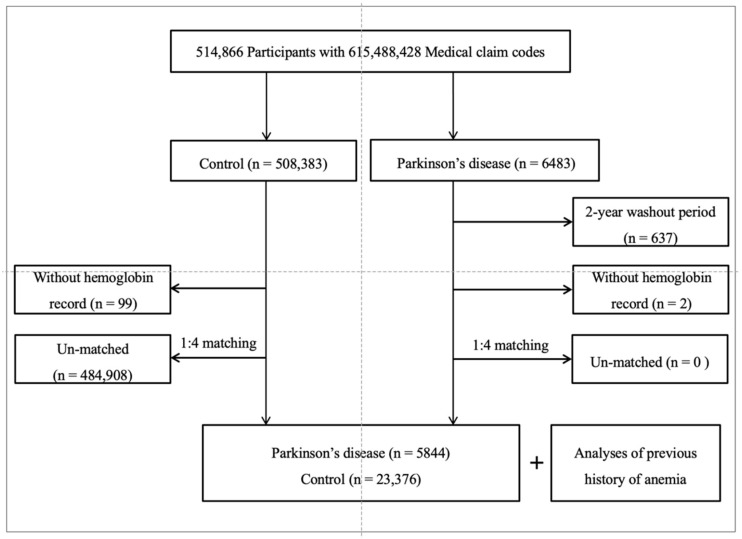
A schematic illustration of the participant selection courses used in this study. Of the total 514,866 participants, 5844 Parkinson’s disease patients were matched with 23,376 control participants for age, sex, income, and region of residence.

**Figure 2 brainsci-11-00623-f002:**
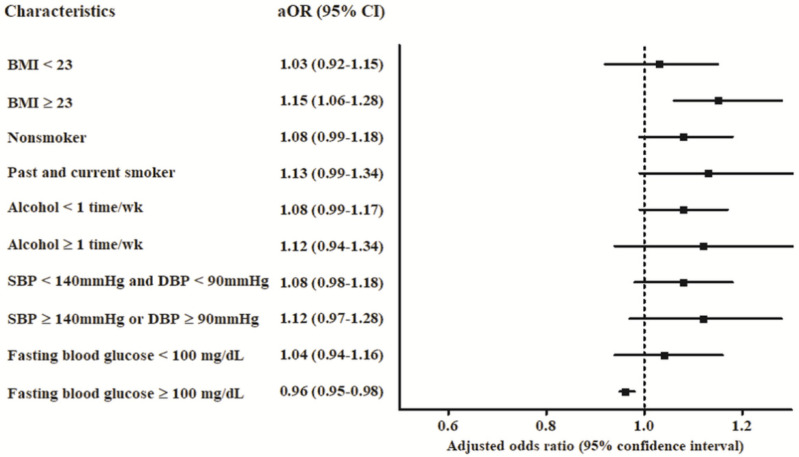
Adjusted odds ratios and 95% confidence intervals were calculated for the risk of PD associated with anemia stratified by comorbidities. aOR–Adjusted odds ratio; BMI–Body mass index; CI–Confidence interval; DBP–Diastolic blood pressure; SBP–Systolic blood pressure.

**Table 1 brainsci-11-00623-t001:** General characteristics of participants.

Characteristics	Total Participants		
	PD (n, %)	Control (n, %)	*p* Value
Age (years old)			1.000
40–44	5 (0.1)	20 (0.1)	
45–49	66 (1.1)	264 (1.1)	
50–54	216 (3.7)	864 (3.7)	
55–59	345 (5.9)	1380 (5.9)	
60–64	612 (10.5)	2448 (10.5)	
65–69	973 (16.7)	3892 (16.7)	
70–74	1369 (23.4)	5476 (23.4)	
75–79	1360 (23.3)	5440 (23.3)	
80–84	716 (12.3)	2864 (12.3)	
85+	182 (3.1)	728 (3.1)	
Sex			1.000
Male	2750 (47.1)	11,000 (47.1)	
Female	3094 (52.9)	12,376 (52.9)	
Income			1.000
1 (lowest)	1111 (19.0)	4444 (19.0)	
2	639 (10.9)	2556 (10.9)	
3	794 (13.6)	3176 (13.6)	
4	1100 (18.8)	4400 (18.8)	
5 (highest)	2200 (37.7)	8800 (37.7)	
Region of residence			1.000
Urban	2185 (37.4)	8740 (37.4)	
Rural	3659 (62.6)	14,636 (62.6)	
Obesity ^b^			0.145
Underweight	239 (4.1)	841 (3.6)	
Normal	2077 (35.5)	8590 (36.8)	
Overweight	1525 (26.1)	6150 (26.3)	
Obese I	1819 (31.1)	7122 (30.5)	
Obese II	184 (3.2)	673 (2.9)	
Smoking status			<0.001 ^a^
Nonsmoker	4599 (78.7)	17,579 (75.2)	
Past smoker	653 (11.2)	2824 (12.1)	
Current smoker	592 (10.1)	2973 (12.7)	
Alcohol consumption			
<1 time a week	4545 (77.8)	16,956 (72.5)	<0.001 ^a^
≥1 time a week	1299 (22.2)	6420 (27.5)	
Systolic blood pressure			0.821
<120 mmHg	1333 (22.8)	5311 (22.7)	
120–139 mmHg	2788 (47.7)	11,254 (48.1)	
≥140 mmHg	1723 (29.5)	6811 (29.1)	
Diastolic blood pressure			0.998
<80 mmHg	2583 (44.2)	10,326 (44.2)	
80–89 mmHg	2107 (36.1)	8437 (36.1)	
≥90 mmHg	1154 (19.8)	4613 (19.7)	
Fasting blood glucose			<0.001 ^a^
<100 mg/dL	3149 (53.9)	13,671 (58.5)	
100–125 mg/dL	1856 (31.8)	7164 (30.7)	
≥126 mg/dL	839 (14.4)	2541 (10.9)	
CCI score			<0.001 ^a^
0	2889 (49.4)	14,632 (62.6)	
1	1117 (19.1)	3561 (15.2)	
2	874 (15.0)	2476 (10.6)	
3	448 (7.7)	1224 (5.2)	
≥4	516 (8.8)	1483 (6.3)	
Other degenerative diseases of the nervous system	464 (7.9)	517 (2.2)	<0.001 ^a^
Head trauma history	460 (7.9)	828 (3.5)	<0.001 ^a^
Anemia	1090 (18.7)	3944 (16.9)	0.001 ^a^

Note: CCI—Charlson comorbidity index; PD—Parkinson’s disease. ^a^ Chi-squared test. Significance at *p* < 0.05; ^b^ Obesity (BMI, body mass index, kg/m^2^) was categorized as <18.5 (underweight), ≥18.5 to <23 (normal), ≥23 to <25 (overweight), ≥25 to <30 (obese I), and ≥30 (obese II).

**Table 2 brainsci-11-00623-t002:** Unadjusted and adjusted odds ratios (95% confidence intervals) for anemia in the Parkinson’s disease and control groups according to age and sex.

Characteristics	Odds Ratios for Anemia			
	Unadjusted ^b^	*p* Value	Adjusted ^c^	*p* Value
Total participants (n = 29,220)				
PD	1.13 (1.05–1.23)	0.001 ^a^	1.09 (1.01–1.18)	0.030 ^a^
Control	1.00		1.00	
Age <70 years old, men (n = 8545)				
PD	1.44 (1.22–1.70)	<0.001 ^a^	1.34 (1.13–1.60)	0.001 ^a^
Control	1.00		1.00	
Age <70 years old, women (n = 9385)				
PD	1.05 (0.92–1.21)	0.468	1.01 (0.87–1.16)	0.924
Control	1.00		1.00	
Age ≥70 years old, men (n = 5205)				
PD	1.14 (0.97–1.34)	0.110	1.09 (0.92–1.28)	0.318
Control	1.00		1.00	
Age ≥70 years old, women (n = 6085)				
PD	1.04 (0.90–1.20)	0.617	1.04 (0.90–1.20)	0.630
Control	1.00		1.00	

Note: CCI–Charlson comorbidity index; PD–Parkinson’s disease. ^a^ Conditional logistic regression model. Significance at *p* < 0.05; ^b^ Models were stratified by age, sex, income, and region of residence; ^c^ Models adjusted for obesity, smoking, alcohol consumption, other degenerative diseases of the nervous system, head trauma history, systolic blood pressure, diastolic blood pressure, fasting blood glucose, and CCI scores.

## Data Availability

The current article used a national sample cohort and does not involve data that can be made available.

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
