# Peer review of "The Association between Anemia and Parkinson’s Disease: A Nested Case-Control Study Using a National Health Screening Cohort"

_brainsci, 2021, doi:10.3390/brainsci11050623_

Round 1

Reviewer 1 Report

This is a decent case control study with very well data processing and detailed description of statistics. The consideration of all potential comorbidities for PD is reasonable and the conclusion based on the analysis is also reasonable. One thing the authors can do better is to do further sub-grouping the anemia or sub-typing the anemia and analyze the relationship with PD. But due to the limitation of the existing data, I understood that the author could not do much changes. 

Reviewer 2 Report

I believe that the study performed is important. Anemia could definitely be a comorbidity in Parkinson's disease.
